# O3D: Offline Data-driven Discovery and Distillation for Sequential Decision-Making with Large Language models

## Abstract

Recent advancements in large language models (LLMs) have exhibited promising performance in solving sequential decision-making problems. By imitating few-shot examples provided in the prompts (i.e., in-context learning), an LLM agent can interact with an external environment and complete given tasks without additional training. However, such few-shot examples are often insufficient to generate high quality solutions for complex and long-horizon tasks, while the limited context length cannot consume larger-scale demonstrations. To this end, we propose an offline learning framework that utilizes offline data at scale (e.g, logs of human interactions) to facilitate the in-context learning performance of LLM agents. We formally define LLM-powered policies with both text-based approaches and code-based approaches. We then introduce *an Offline Data-driven Discovery and Distillation* (O3D) framework to improve LLM-powered policies without finetuning. O3D automatically discovers reusable skills and distills generalizable knowledge across multiple tasks based on offline interaction data, advancing the capability of solving downstream tasks. Empirical results under two interactive decision-making benchmarks (ALFWorld and WebShop) demonstrate that O3D can notably enhance the decision-making capabilities of LLMs through the offline discovery and distillation process, and consistently outperform baselines across various LLMs with both text-based-policy and code-based-policy.

## 1 Introduction

Recent years have witnessed remarkable advancements in artificial intelligence (AI), particularly in the development of Large Language Models (LLMs). One of the standout features of LLMs is their in-context learning ability, where the LLM can perform tasks with only a few-shot examples provided in the prompts, making it possible to deploy LLMs to various applications seamlessly.

Although most existing research focuses on one-step text generation such as question answering, many real-world scenarios desire autonomous agents that can interact with external environments and make sequential decisions to complete given tasks. There are some recent works that successfully showcase the application of LLMs in sequential decision-making (Yao et al., 2023b; Shinn et al., 2023; Liu et al., 2023c; Yang et al., 2023), by either directly letting the language model interact with the environment, or using LLMs to write code which then executes in the environment. With a few examples of acting and reasoning (Yao et al., 2023b), an LLM-based agent can interact with the environment and even learn by *reflecting* on historical errors (Shinn et al., 2023).

However, existing methods still struggle to solve many complex domains with LLMs due to the intrinsic difficulties that arise from long-horizon interactive tasks. On the one hand, it is widely known that the task complexity increases exponentially with the interaction horizon (Sutton & Barto, 2018), such that a large amount of data or demonstrations can be desired for an agent to fully understand the environment dynamics, especially for heterogeneous real-world environments and tasks, where cross-task generalizability is important. On the other hand, the in-context learning ability is constrained by the limited context window of an LLM. Even if many demonstrations exist, it is hard to prompt LLMs with sufficient examples. Although finetuning is a solution, it can be much more expensive and less accessible for normal users.

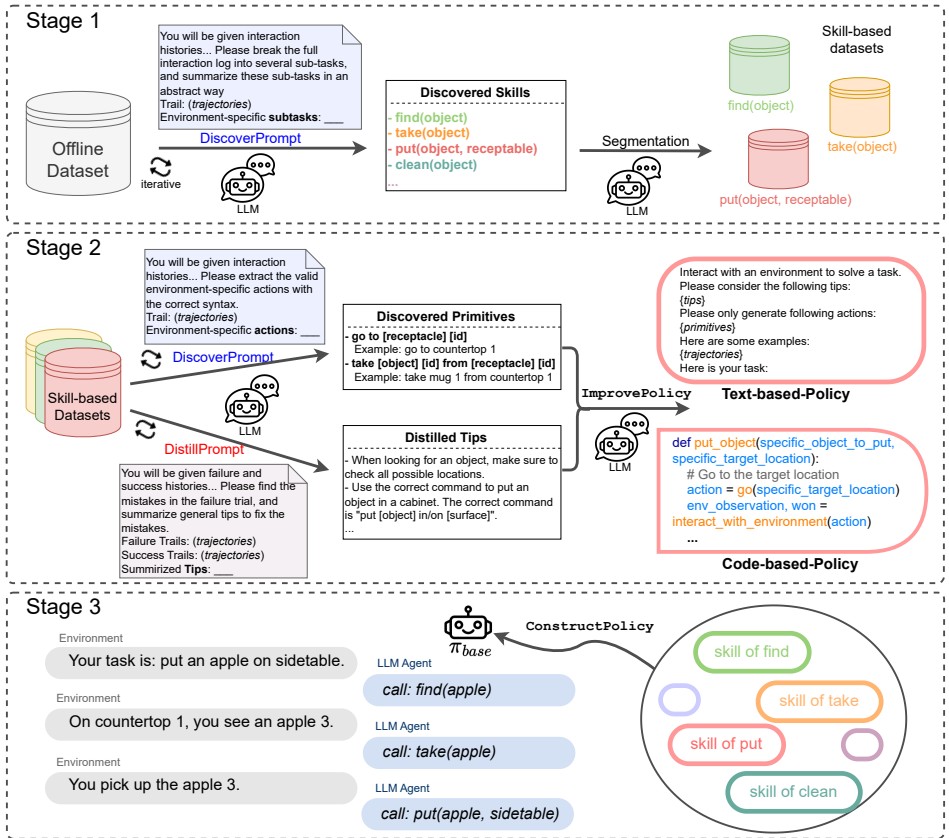

Figure 1: The proposed O3D framework.

In response to these challenges, this paper asks and aims to answer the follow question:

*Can we develop a data-driven learning framework for sequential decision-making with LLMs, such that LLMs can learn from large-scale offline data without the need of model training?*

In this paper, we define an offline learning framework to enable LLMs to discover and distill useful knowledge from interaction trajectories on multiple tasks. We first formalize LLM-powered policies that unify the two parallel approaches in literature which generate text and code as policies, respectively. For these LLM-powered policies, we carefully design a generic learning paradigm called *Offline Data-driven Discovery and Distillation* (O3D), which can iterate over the offline dataset and keep improving the policy. Importantly, our method does not require a high-quality expert offline dataset, as it can benefit from both positive examples and negative examples of environment interactions, making the framework easier and cheaper to use in practice. As shown in Figure 1, O3D is composed of 3 stages: the first stage aims to discover reusable skills by segmenting the offline interaction trajectories; the second stage then conducts skill-conditioned policy improvement by distilling knowledge from offline data; the third stage constructs the interactive policy by calling the learned skills given diverse tasks. All stages are based on querying LLMs and iterating over existing text- or code-based policies. As a result, O3D can learn better policies from offline dataset at scale without any model finetuning. Experiments in two commonly used domains (ALFWorld and WebShop) show that our LLM agent augmented by offline knowledge has much better few-shot performance than prior methods on a variety of downstream tasks.

**Summary of contributions: (1)** We establish the first offline in-context learning framework for LLM sequential decision-making agents, such that the LLM can learn from offline experience without any finetuning. This offline learning paradigm allows more effective usage of past interactions (including both good and bad behaviors) generated by human or other agents, alleviating the cost of online learning. **(2)** Our offline learning algorithm unifies two common approaches of LLM-based decision-making: textual action generation and code generation. For both approaches, our method achieves significant improvement over baseline methods on challenging domains. **(3)** Different

from prior work which prompts and solves different types of tasks independently, our algorithm can leverage offline experience from multiple tasks. By letting LLMs automatically distill shared high-level knowledge, our algorithm achieves few-shot generalization to various types of tasks with a single set of prompts.

## 2 RELATED WORK

**LLMs for sequential decision-making with in-context learning.**

• *Text-based methods.* Yang et al. (2023) and Liu et al. (2023b) conduct extensive experiments to showcase the ability of LLMs to make sequential decisions in a range of challenging multi-step reasoning tasks. Yao et al. (2023b) propose ReAct, a method to combine multi-step reasoning and interactive decision-making, which achieves significant improvement in multiple domains. Shinn et al. (2023) develop agents that can verbally summarize from its past failures and incorporate the reflective text into subsequent trials of the same task, analogous to an online reinforcement learning paradigm. Our method, in contrast, adopts an offline learning method to minimize the cost of online learning and can adapt to various new tasks.

• *Code-based methods.* It has been shown in multiple domains that embodied agents can be empowered by LLM-written code Liang et al. (2022); Wang et al. (2023a). Sun et al. (2023) propose a method to refine the code-based plan during interaction rather than executing in a fixed loop. Liu et al. (2023c) propose to extract hierarchical summary from robot past experiences to adjust the plan. A concurrent work by Wang et al. (2023b) proposes a framework that generates robot task code from demonstrations via recursive task summarization. But our O3D differs from these methods as 1) O3D can iteratively improve the policy by utilizing offline data at scale, 2) O3D takes a bottom-up approach rather than a top-down recursive way to decompose the task, and 3) O3D works for both text-based policies and code-based policies.

• *Combining LLMs with existing planning approaches.* It has also been shown that LLMs can be combined with classical planning approaches, such as Planning Domain Definition Language (PDDL) (Liu et al., 2023a; Silver et al., 2023). Differently, our paper focuses on end-to-end LLM policies without additional planning algorithm or knowledge.

**Training or fine-tuning LLMs for sequential decision-making.** Another line of related work includes training textual policies with imitation learning (Shridhar et al., 2021; 2020) and fine-tuning language models to behave as policies (Wang et al., 2023a; Driess et al., 2023; Wang et al., 2023c) in sequential decision-making. Again, our work is different as it aims at achieving high-quality LLM-based decision-making without fine-tuning.

**Multi-step reasoning and task decomposition with LLMs.** Multi-step reasoning using LLMs has been widely studied in language domains, including chain-of-thought style reasoning Yao et al. (2023a); Fu et al. (2022); Wei et al. (2022), step-by-step feedback based reasoning Lightman et al. (2023); Zheng et al. (2023), and self consistency and verification based reasoning Ling et al. (2023); Wang et al. (2022). In contrast, we focus on sequential decision-making tasks in partially observable environments, where each step results in state transitions and only sparse rewards are available.

## 3 METHODOLOGY

### 3.1 PROBLEM FORMULATION

**LLM-powered Policy.** We first formally define an LLM-powered policy. A policy $\pi$ for sequential decision-making is a function that maps the interaction history to a distribution over actions. Let $\pi(a|\tau)$ denote the probability of selection action $a$ given interaction history $\tau$, which is a sequence of all past observations and actions, $\langle o_1, a_1, o_2, a_2, \cdots, o_t \rangle$. Then, with a pre-trained LLM, a policy can be realized in the following two ways.

• *Text-based-Policy.* With a pre-trained LLM which outputs text based on any text input, a policy can be written as

$$\pi_{\text{text}}(a|\tau) := LLM(a|\tau; \theta_{\text{pmt}}, \theta_{\text{pret}}). \tag{1}$$

• *Code-based-Policy.* It is well-known that LLMs can program, such that one can ask LLM to directly generate code to implement the policy function, i.e.,

$$\pi_{\text{code}}(a|\tau) := Code(a|\tau) \leftarrow LLM(\theta_{\text{pmt}}, \theta_{\text{pret}}). \tag{2}$$

The goal is to learn a policy that can maximize the total reward. In both the above cases, the pre-trained LLM weights $\theta_{\text{pret}}$ are fixed, and our O3D learns a policy by learning and optimizing the base prompt $\theta_{\text{pmt}}$ as well the written policy code $Code(a|\tau)$ from offline data.

**Skill-conditioned Policy.** A policy can be conditioned on specific skills or subgoals (e.g., find a mug), which are compositional factors of the original task (e.g., heat some milk). Let $z$ be a skill, then a skill-conditional policy can be denoted as $\pi^z$, with $\pi^z(a|\tau)$ the probability of selecting action $a$ given history $\tau$ when executing skill $z$.

## 3.2 LEARNING FROM OFFLINE DATA

In many real-world decision-making systems, there exists interaction log from various users, including experts who can successfully perform the task, as well as non-experts who may fail and make mistakes. Our goal is to learn a good "policy" defined in Section 3.1 by utilizing the offline dataset to learn the base prompt, but having the model weights, $\theta_{pret}$, fixed.

Intuitively, seeing the interaction logs from others performing a task can be helpful for one to understand the environment and finish similar tasks. Since LLMs have strong abilities of interpretation and generalization, recent works such as ReAct (Yao et al., 2023b) have shown that LLMs can solve many interactive decision-making problems when prompted with a few expert demonstrations. This learning paradigm is analogous to behavior cloning, where an agent learns to imitate how experts react to certain scenarios. However, traditional behavior cloning suffers from the distribution shift between expert demonstrations and the agent's own online interactions, especially when the expert dataset is small and not representative of all scenarios in the domain. Although LLMs are powerful at interpolating and generalizing with the pre-trained language understanding ability, their fixed context length only allows a limited number of expert demonstrations, making it hard to fully understand an external decision-making environment with specific dynamics and requirements. That is, even when there exist a rich and diverse offline dataset, in-context behavior cloning (Yao et al., 2023b) is only able to utilize a small subset of the data and obtain sub-optimal policies. To overcome this issue, we introduce an offline learning framework for LLM-powered policies, including both Text-based-Policy and Code-based-Policy defined in Section 3.1.

## 3.3 O3D: A FRAMEWORK OF LLM-BASED OFFLINE POLICY IMPROVEMENT

Our proposed offline policy learning framework consists of 3 stages, as depicted in Figure 1. The first stage enables the LLM to discover and abstract reusable skills from offline datasets (potentially from diverse tasks). Then, the second stage aims to learn a skill-based policy for each discovered skill, through iterative discovery and primitives and iterative policy improvement with knowledge distillation. The final stage is to construct the main LLM-powered agent who can reason and call corresponding skills sequentially to solve given tasks. Below we explain each stage in detail.

**Stage 1: Offline Skill Discovery and Data Segmentation.** Many real-world decision-making processes requires a number of steps to complete a task, such as controlling a robot to pass several obstacles and navigate to the door, which results in two challenges for LLM-powered agents. First, the limited context length may not be enough to contain the few-shot demonstration and online interaction history. Second, the language model may lose track of its goal and not pay attention to the most important information. To mitigate this issue, we propose a hierarchical policy learning framework that can iteratively extracts skills from interaction logs with primitive-level executions. Here the skills are analogous to the options or temporally extended actions (Sutton et al., 1999) in hierarchical reinforcement learning. It is well-known that discovering options is difficult in traditional RL, whereas we find that skill discovery with textual logs can be well-achieved with the semantic understanding ability of LLMs.
Our skill discovery process iterates over the offline trajectories, using a DiscoverPrompt as shown in Figure 1 (Stage 1). The full prompt we use is in Appendix B. We ask LLMs to divide the interaction histories into skill-oriented sub-trajectories, and abstract the skills in function forms. For example, from all 6 types of ALFWorld (Shridhar et al., 2021) tasks, the LLM can reliably discover 7 skills: find(object), take(object), clean(object), heat(object), cool(object), use(object) and put(object, receptacle), covering all the required sub-procedures in the domain and are reusable across tasks.

**Stage 2: Offline Policy Improvement with Knowledge Distillation.** The main idea of this stage is to distill generalizable knowledge from offline datasets, such that the knowledge can improve the policy's performance in downstream tasks. Such knowledge should be generalizable to tolerate the distribution shift between offline data and online interactions. We propose to distill the following types of knowledge from the segmented skill-based trajectories in an iterative process, which leads to improved skill-conditioned policies.

• *Distilling Primitive Actions.* A common mistake of LLM-powered agents is hallucination, i.e., the agent outputs actions that are not valid in the environment. To ensure effective usage of LLM in decision-making applications, it is important to specify the space of actions in the form of natural language or code. Many prior works (Liang et al., 2022; Liu et al., 2023c) manually define the available primitive functions, which requires human labor and domain knowledge. Instead, we propose to distill primitive actions or functions from the offline interaction data with LLM, which is easy to scale up and automate the practical operation. Figure 1 describes how to distill the primitives with an example, and the full prompt is in Appendix B.

• *Distilling Policy Improvement Tips with Trajectory Contrasting.* Inspired by the policy gradient methods in RL, which increases the probability of selecting good actions and lower the probability of selecting bad ones, we propose to distill knowledge that can enhance good (i.e., can incur high long-term reward) behaviors and avoid undesired behaviors in the task distribution. We propose to distill "policy improvement tips" about what actions are preferred under what circumstances. However, with the offline data that only provides sequences of interactions and final scores, it is non-trivial for an LLM to figure out the correct credit assignment and the useful tips to guide policy improvement. To this end, we propose *Trajectory Contrasting*, where we sample both successful (high-score) and failed (low-score) trajectories and let the LLM generate tips by contrasting them. As a result, the LLM can identify the key to success and how to avoid failures. Figure 1 shows the distillation process with a DistillPrompt which iteratively updates an LLM-powered policy. More implementation details for Text-based-Policy and Code-based-Policy are provided in Section 3.4.

**Stage 3: Downstream Interaction with Hierarchical Policy Execution.** So far, Stage 1 discovers a set of skills, while Stage 2 produces and optimizes the corresponding skill-conditioned policies. The final stage is then to compose these policies and interact with the downstream task by calling the learned policies. We prompt a base policy $\pi_{\text{base}}$ with a few examples (come from LLM's own segmentation of offline trajectories) on calling the proper skills sequentially given a downstream task. Figure 2 shows an example of how to construct the base policy by prompting and how a skill-conditioned policy is prompted when being called.

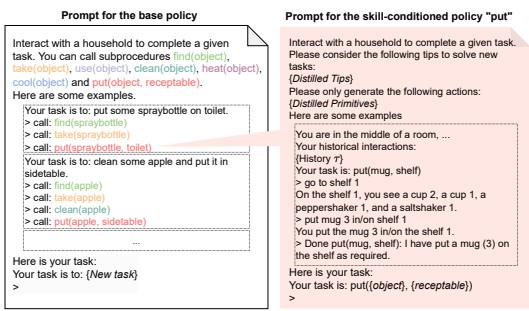

Figure 2: Example prompts for the base policy and the text-based skill-conditioned policy for hierarchical policy execution in Stage 3.

### 3.4 IMPLEMENTATION DETAILS OF O3D

The main algorithm is presented in Algorithm 1, and the implementation-specific functions for Text-based-Policy and Code-based-Policy are defined in Algorithm 2 and Algorithm 3, respectively. We provide all used prompts and additional implementation details in Appendix B. The major **difference in implementation** between Text-based-Policy and Code-based-Policy is in the policy formulation and improvement processes:

• *Policy Initilization with Primitives.* Text-based-Policy directly provides the discovered primitives in the prompt of policy and advises the agent to follow the primitives, while Code-based-Policy first lets the LLM write primitive functions and then calls these primitive functions in the code of skill-conditioned policies.

• *Policy Improvement.* Since the distilled policy improvement tips are in natural language, we directly ask the LLM to merge the new suggestion into the prompt of Text-based-Policy. For Code-based-Policy, we let the LLM consider the policy improvement tips and re-write the policy code.

• *Policy Construction and Execution.* In Stage 3, we prompt the base policy to call the learned Text-based-Policy or Code-based-Policy. Note that for Code-based-Policy, it is possible that the generated skill-conditioned code has compilation errors, so that it requires human checking or validation on a small set of tasks to verify that the code is executable.

---

**Algorithm 1:** Policy Learning with Offline Data-driven Discovery and Distillation (O3D)

---

1 **Input:** Pre-trained LLM, offline dataset $\mathcal{D}$, batch sizes $N_1, N_2$, max iteration steps $T_1, T_2$
2 **Output:** LLM-powered policy $\pi$
   // Stage 1: skill discovery and data segmentation
3 Initialize sets of skills $\mathcal{Z} = \emptyset$
4 **for** $t = 1, \ldots, T_1$ **do**
5    Sample $N_1$ trajectories $d \sim \mathcal{D}$
6    Attempt to discover more skills $\mathcal{Z} \leftarrow LLM(\mathcal{Z}, d; \text{\color{blue}DiscoverPrompt})$
7 Segment trajectories in $\mathcal{D}$ based on skillset $\mathcal{Z}$, obtain $\mathcal{D}^{z_k}$ for each $z_k \in \mathcal{Z}$
   // Stage 2: skill-conditioned policy improvement
8 **for** $z_k \in \mathcal{Z}$ **do**
9    Initialize the primitive set $\mathcal{P}^{z_k}$ and the knowledge set $\mathcal{T}^{z_k}$
10    Initialize $\pi^{z_k} \leftarrow \text{InitSkillPolicy}(\mathcal{D}^{z_k}, \mathcal{P}^{z_k})$
11    **for** $t = 1, \ldots, T_2$ **do**
12       Sample $N_2$ trajectories $d^{z_k} \sim \mathcal{D}^{z_k}$
13       Attempt to discover more primitives $\mathcal{P}^{z_k} \leftarrow LLM(\mathcal{P}^{z_k}, d^{z_k}; \text{\color{blue}DiscoverPrompt})$
14       Distill policy improvement knowledge $\mathcal{T}^{z_k} \leftarrow LLM(\mathcal{T}^{z_k}, d^{z_k}; \text{\color{red}DistillPrompt})$
15       Improve the policy with procedure $\text{ImprovePolicy}(\mathcal{T}^{z_k}, \pi^{z_k})$
   // Stage 3: policy composition and downstream interaction
16 Construct main policy $\pi \leftarrow \text{ConstructPolicy}(\mathcal{D}, \mathcal{Z})$ and interact with downstream tasks

---

---

**Algorithm 2:** Text-based-Policy

---

1 **Function** $\text{InitSkillPolicy}(\mathcal{D}^z, \mathcal{P}^z)$**:**
2    Sample examples $d \sim \mathcal{D}^z$
3    Initiate $\theta_{\text{pmt}}$ with $d$ and $\mathcal{P}^z$
4    **return** $LLM(\theta_{\text{pmt}}, \theta_{\text{pret}})$ as policy
5 **Function** $\text{ImprovePolicy}(\mathcal{T}^z, \pi^z)$**:**
6    Ask LLM to incorporate $\mathcal{T}^z$ into the prompt of policy $\pi^z$
7 **Function** $\text{ConstructPolicy}(\mathcal{D}, \mathcal{Z})$**:**
8    Sample examples $d \sim \mathcal{D}$ and segment them based on skills
9    Provide the examples as demonstrations for $\pi_{\text{text}}$ to call skills given the task
10    **return** Text-based-Policy $\pi_{\text{text}}$

---

**Algorithm 3:** Code-based-Policy

---

1 **Function** $\text{InitSkillPolicy}(\mathcal{D}^z, \mathcal{P}^z)$**:**
2    Sample examples $d \sim \mathcal{D}^z$
3    Let LLM write a function to reproduce $d$ with primitive functions $\mathcal{P}^z$
4    **return** generated $Code$ as policy
5 **Function** $\text{ImprovePolicy}(\mathcal{T}^z, \pi^z)$**:**
6    Let LLM improve the code $\pi^z$ based on $\mathcal{T}^z$
7 **Function** $\text{ConstructPolicy}(\mathcal{D}, \mathcal{Z})$**:**
8    Sample examples $d \sim \mathcal{D}$ and segment them based on skills
9    Let LLM write code as $\pi_{\text{code}}$ to call skills given the task
10    **return** Code-based-Policy $\pi_{\text{code}}$

---

Using LLMs to directly interact with environments (Text-based-Policy) and using LLMs to write code to interact with environments (Code-based-Policy) are usually discussed separately. In this paper, we take the first step to unify and compare these two approaches in a single framework. Our study also reveals the different **pros and cons** of these two approaches.

• *Advantages of Code-based-Policy.* Code-based-Policy explicitly writes the acting policy in code, which is more interpretable and reliable, and can fully avoid hallucination or syntax errors in execution. Moreover, Code-based-Policy is usually more cost efficient, as the generated code can be reused in new tasks without calling LLMs. Therefore, Code-based-Policy can be more suitable for applications where reliability and efficiency are important.

• *Advantages of Text-based-Policy.* Text-based-Policy is relatively easy to implement in practice with less human supervision. Also, in complicated environments such as WebShop where language understanding is important, Text-based-Policy can achieve much better performance than Code-based-Policy, as it retains the commonsense, expressiveness and reasoning ability of pre-trained LLMs. Therefore, for language-oriented applications where reasoning and the ability of recovering from failure are crucial, Text-based-Policy, or a combination of the two approaches, can be a better choice.

## 4 EXPERIMENTS

### 4.1 EXPERIMENTAL SETUP

**Problem Domains.** We consider two sequential decision-making benchmarks, ALFWorld (Shridhar et al., 2021) and WebShop (Yao et al., 2022). ALFWorld is a unique environment that mimics

Table 1: Results in ALFWorld

| Model | Method | Pick | Clean | Heat | Cool | Look | Pick2 | All |
|-------|--------|------|-------|------|------|------|-------|-----|
| | | Text-based Policy | | | | | | |
| GPT-4 (0613) | ReAct | 67 | 74 | 74 | 67 | 100 | 47 | 72 |
| | O3D | **92** | **100** | **96** | **95** | **100** | **53** | **91** |
| | O3D-Human | 83 | 100 | 87 | 95 | 100 | 53 | 88 |
| GPT-3.5 (0613) | ReAct | 13 | 10 | 0 | 0 | 17 | 0 | 7 |
| | O3D | 71 | 35 | 4 | 67 | 44 | 24 | 41 |
| | O3D-Human | 71 | 68 | 83 | 71 | 44 | 24 | 63 |
| GPT-3.5 (0301) | ReAct | 42 | 52 | **65** | 38 | 61 | **29** | 49 |
| | O3D | **92** | 71 | 52 | **57** | **72** | 6 | **61** |
| | O3D-Human | 83 | 87 | 78 | 90 | 44 | 18 | 71 |
| | | Code-based Policy | | | | | | |
| GPT-4 (0613) | Demo2Code | 96 | 58 | 13 | 43 | 0 | 65 | 48 |
| | O3D-Code | **100** | **84** | **87** | **90** | **89** | **88** | **90** |
| GPT-3.5 (0613) | Demo2Code | 96 | 26 | 48 | 29 | 0 | **82** | 46 |
| | O3D-Code | **100** | **71** | **91** | **86** | **89** | 18 | **78** |

Table 2: Results in WebShop

| Model | Method | SR | Score |
|-------|--------|-----|-------|
| | | Text-based Policy | |
| GPT-4 (0613) | ReAct | 26 | 39 |
| | O3D | **41** | **58** |
| | O3D-Human | 41 | 61 |
| GPT-3.5 (0613) | ReAct | 27 | 60 |
| | O3D | 35 | **61** |
| | O3D-Human | 31 | 61 |
| GPT-3.5 (0301) | ReAct | 12 | 28 |
| | O3D | **18** | **33** |
| | O3D-Human | 20 | 35 |
| | | Code-based Policy | |
| GPT-4 (0613) | Demo2Code | 1 | 5 |
| | O3D-Code | **19** | **31** |
| GPT-3.5 (0613) | Demo2Code | 0 | 0 |
| | O3D-Code | 0 | 0 |

household scenarios and allows an decision-making agent to interact with the environment through a text-based interface We use the same test set as introduced in ALFWorld paper, including 134 tasks in total across six distinct task types. Following Shinn et al. (2023), we make the problem even more challenging by limiting the horizon of each episode to be 30 (original is 50) steps and terminating the episode if the agent takes the same action twice. WebShop provides a real-world online shopping environment with 1.18M products, where an agent must explore the website, check relevant product candidates, and purchase the one that matches a user instruction (e.g., "I am looking for a queen sized bed that is black, and price lower than 140.00 dollars"). Our evaluation considers the first 500 out of 12, 087 instructions as test set (following the official implementation (Yao et al., 2022)).

**Baselines and Metrics.** We mainly compare the proposed O3D and O3D-Code with a popular text-based baseline method, ReAct (Yao et al., 2023b) and a state-of-the-art code-based baseline approach, Demo2code (Wang et al., 2023b). Meanwhile, we have a variant of our method, named as O3D-Human, using the knowledge summarized by a human from the logs of ReAct in the test set of each domain, which we treat as an oracle baseline. This design is to investigate if the improvement tips distilled by LLMs can be as effective as the tips distilled by humans. In ALFWorld, we assess method performance by measuring the success rate (SR) under each task type as well as a total success rate over 134 tasks. Besides the success rate, in WebShop, there is a product matching score as an extra metric.

**Models and Offline Data.** To investigate the robustness of O3D across various LLMs, we consider three GPT models (providing different $\theta_{\text{pret}}$ defined in Equation (1) and Equation (2)) in our experiments, including GPT-4-0613, GPT-3.5-0613 and GPT-3.5-0301. The offline data include official human demonstrations in both domains as the success data, and a set of failure data generated by using ReAct on the training task set introduced in the original ALFWorld and WebShop implementations (more details are referred to Appendix A.1).

## 4.2 RESULTS AND ANALYSIS

**Effectiveness of discovering skills and primitives, and distilling policy improvement tips.** In our experiments, O3D efficiently extracts high-quality skills from raw human demonstrations in the offline data, resulting in seven types of skills under ALFWorld domain, including: *find(object)*, *take(object)*, *put(object, receptacle)*, *cool(object)*, *heat(object)*, *use(object)* and *clean(object)*; and four types of skills under WebShop domain, including *search item*, *select item*, *select item's attributes* and *purchase item*. Each skill consists of a set of primitives to execute, along with a group of tips to assist with the skill completion. Fig 3 shows several skill examples in ALFWorld domain (full results are available in Appendix B.3). By learning from offline data, O3D is able to capture correct primitives that can be composed to achieve each corresponding skill. Furthermore, we note that tips distilled by O3D is functional to various degrees, such as suggesting general tips, encouraging exploration, realizing action preconditions and highlighting syntax (as shown in Fig. 3).

**O3D consistently outperforms baselines in both Text-based-Policy and Code-based-Policy across various LLMs under ALFWorld** (Table. 1) and **WebShop** (Table. 2). When using LLMs as text-

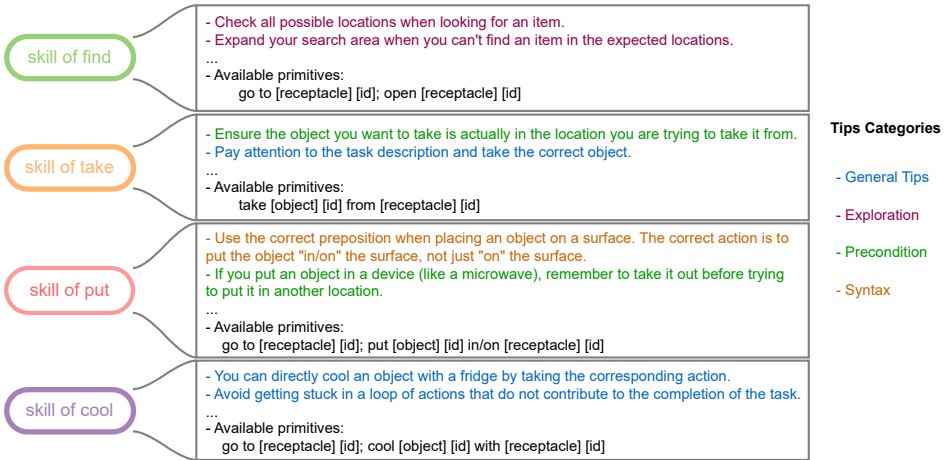

Figure 3: Examples of discovered skills with primitives and distilled knowledge under ALFWorld.

based policies, O3D respectively outperforms ReAct by 19%, 34% and 12% in ALFWorld, and 15%, 8% and 6% in WebShop in terms of success rate, with using GPT-4-0618, GPT-3.5-0613 and GPT-3.5-0301 repsectively. Furthermore, as shown in Table 1, the success rate achieved by O3D under each task category is always greater than the one achieved by ReAct with GPT-4-0613 and GPT-3.5-0613. This further confirms that the tips distilled by O3D from offline data is generalizable and useful across diverse task types. For example, the tip of "pay attention to the task description and take the correct object" helps the LLM agent avoid taking a similar object (a pot) rather than the requested one (a pan), and the tip of "The correct action is to put the object 'in/on' the surface, not just 'on' the surface" prevents the LLM agent from using wrong syntax of primitives, which are the two common mistakes made by ReAct). Importantly, O3D can achieve competitive performance with O3D-Human in both domains, and even surpasses O3D-Human when using GPT-4 in ALFWorld and GPT-3.5-0613 in WebShop. This is strong evidence to validate that O3D's knowledge distillation is a promising approach to extract human-level tips from offline data to enhance the capability of an LLM to solve downstream tasks without finetuning or labor-intensive prompt engineering.

In ALFWorld, O3D-Code surpasses Demo2Code, achieving a remarkable 32% higher performance with GPT-4-0613 and 14% with GPT-3.5-0613. Additionally, there's an 18% performance improvement on WebShop tasks using GPT-4-0613. However, GPT-3.5 struggles with these tasks due to the intricate nature of language processing and the complexity of the logic in WebShop. The principal advantage of our approach lies in its unique method of generating code: it adopts a bottom-up style, effectively constructing policies on top of robust skill functions. Through iterative skill refinement, the model cultivates robust skills by leveraging extensive and diverse data from demonstrations. Skill functions can then be efficiently utilized and reused to compose higher-level policies. In contrast, Demo2Code follows a top-down approach, requiring the generation of code for the same set of skills each time it receives a new task. Due to the context length constraint inherent in LLMs, only a limited number of demonstrations are used to guide skill code generation in Demo2Code, resulting in unstable and inconsistent skills.

Webshop poses a substantial challenge for code-based policies owing to its need for comprehensive natural language understanding within the environment feedback. We address this challenge by enabling LLMs to construct skills by employing a limited set of LLM-based functions that encapsulate the underlying LLM capabilities (see Appendix B.5.2). To ensure a fair comparison, we also offer the same set of functions to Demo2Code. While our approach substantially enhances the performance of code-based policies in comparison to the baseline, it's important to note that skill generation remains considerably constrained by the intricate and diverse text-based environment feedback.

**Primitives, skills and policy improvement tips independently advance baseline performance in both ALFWorld and WebShop.** O3D has three major processes: skill discovery (SD), primitives discovery (PD) and policy improvement tip distillation (TD). To investigate each component's contribution to performance in downstream task solving, we conducted an ablation study considering three variants of O3D, each with only one component. Fig. 4 shows that, in both domains, the three variants of O3D either outperform the baseline or achieve the same performance as the baseline across tested

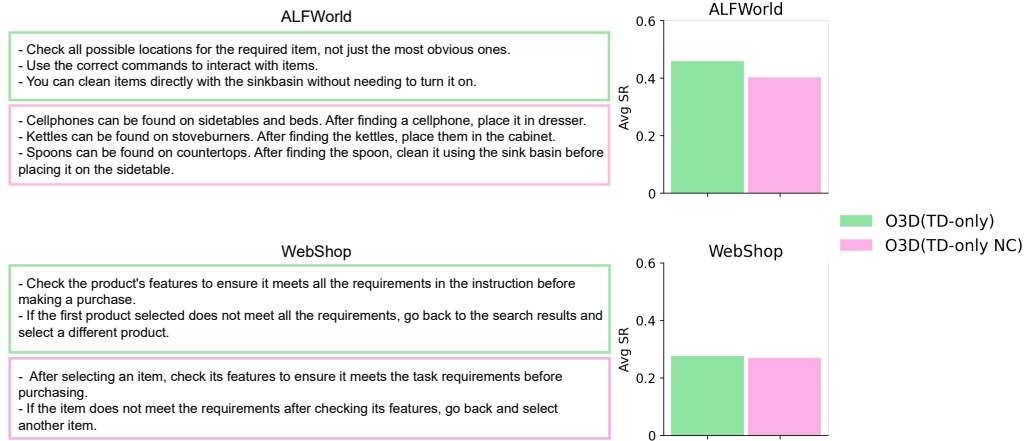

Figure 4: Comparison on success rate (SR) with three variants of O3D against the baseline method.

LLM models. In ALFWorld, O3D (PD-only) plays the dominant role in performance improvement with GPT-4-0613 and GPT-3.5-0301. This is because the major mistakes made by the baseline are in terms of outputting primitive actions with syntax errors or hallucinating unavailable actions. O3D (SD-only) boosts the performance the most with GPT-3.5-0613, becuase the tasks in ALFWorld are too complex for ReAct with GPT-3.5-0613, and O3D (SD-only) solves the tasks in hierarchy by performing skill selections that greatly reduces the complexity. In WebShop, the three components consistently benefit the baseline performance across the three GPT models, with their individual contributions also being model-dependent. Since the offline data was collected using GPT-3.5-0613, we observe that the highest overall improvement of the three components occurs in this model.

Figure 5: Comparison on averaged success rate over GPT models between using tips distilled via contrastive and non-contrastive (NC) methods, with examples in green and pink boxes respectively.

**The advantage of using a contrastive method to distill improvement tips versus a non-contrastive method is domain-dependent.** Fig. 5 shows that the proposed trajectory contrasting, which compares both successful and failed trials in offline data, is relatively helpful in certain domains, compared with the non-contrastive (NC) way based on only success data. In ALFWorld, failures in baseline method are often caused by violations of the domain-specific dynamics and rules. Therefore, the contrastive approach can generate general tips (green box in Fig. 5) to correct mistakes that occur in failure cases, while the non-contrastive approach only summarizes the facts (pink box in Fig. 5) from successful trials, which is less helpful. However, in WebShop, the two approaches achieve similar performance, as they output very analogous tips as shown in the corresponding boxes in Fig. 5.

## 5 CONCLUSION

This paper introduces an offline in-context learning framework, O3D, for LLM sequential decision-making agents, where agents can learn from previous experiences in a scalable offline manner to improve performance without the need of fine-tuning. O3D stands out by allowing LLMs to distill shared high-level knowledge from offline interaction logs, which is injected into a single set of prompts to be reused in solving diverse downstream tasks. Empirically, O3D leverages a unified algorithm that enhances both text-based policies and code-based policies with LLMs, outperforming baseline methods in two challenging benchmark domains. Our work offers a possibility for future LLM-based algorithm development in terms of efficiently leveraging offline data at scale for real-world sequential decision-making applications.

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
