# OpenReview forum: "O3D: Offline Data-driven Discovery and Distillation for Sequential Decision-Making with Large Language Models"
_ICLR.cc/2024/Conference — Submitted to ICLR 2024_

### Official Review · Reviewer_x6tG · 2023-11-03

**Soundness:** 2 fair
**Presentation:** 3 good
**Contribution:** 2 fair
**Rating:** 3
**Confidence:** 4

**Summary:**

This paper provides O3D (Offline Data-driven Discovery and Distillation), a new prompting method for Large Language Model (LLM) agents to solve decision-making tasks. The main idea of O3D is to use offline trajectory data to improve a base prompt for a LLM agent. To do this, the paper reformulates a LLM-based policy p(a | tau) as a LLM-based policy parameterized by prompts LLM(a | tau; prompt). More specifically, O3D discovers primitive skills from offline trajectory data by using a LLM, and construct a skill-conditioned prompt. Then, O3 distills improvement tips from offline trajectory data by using a LLM, and improvs the base prompt by conditioning the distilled tips. In summary, O3D consists of three stages: (1) offline skill discovery and data segmentation, (2) offline policy improvement with distilled improvement tips, and (3) hierarchical policy execution. Providing experiment results on ALFWorld and WeShop, this paper shows that O3D can provides better success rate than ReAct, a recent representative method.

**Strengths:**

- S1. The proposed idea of improving a base prompt for a LLM agent by using offline trajectory data is interesting. Especially, it is interesting to use a pair of high/low reward trajectories to automatically generates improvement tips by using a LLM.

- S2. This paper provides two alternative methods: (1) text-based policy and (2) code-based policy. Also, it provides a comparison between them.

**Weaknesses:**

- W1. Since this paper combines many techniques such as skill-conditioning in a prompt, prompt refinements by improvement tips, and code-based policy, it is rather hard what is the main contribution. If the main contribution is to improve a base prompt by using textual tips, how does O3D differ from recent works like Reflexion?

- W2. One of main concerns on this paper is experiment. Since this paper compares O3D with only one baseline (i.g., ReAct for text-based policy and Demo2Cod for code-based policy), it is hard to properly assess the performance. For example, there are more recent works such as Reflexion for text-based policy. It is highly required to add more recent works and properly compare O3D with them. Also, the ReAct paper provides its performance by using PaLM and text-davinci-002 (GPT-3). However, this paper provides results based on GPT-4 and GPT-3.5. This mismatch makes the comparison more difficult.

- W3. According to Table 1 and 2, code-based policy does not work well in WebShop. This result does not seem to support that O3D consistently outperforms baselines in both text-based policy and code-based policy across various LLMs.

- W4. The proposed method seems more suitable for ALFWorld than WebShop. I am not sure that O3D is generally applicable to diverse environments.

**Questions:**

- Q1. Regarding W1, how does O3D differ from self-refinement works like Reflexion?

- Q2. Regarding W3, how does the skill discovery work for WebShop?

- Q3. What are the details of O3D-Human?

---

> ### Author Response · Authors · 2023-11-21
> **[1 / 3] Response to Reviewer x6tG**
>
> We greatly appreciate the valuable feedback from Reviewer x6tG. We are encouraged that the reviewer thinks that our proposed learning from offline data and contrastive knowledge distillation are interesting ideas. Below we address the concerns and questions raised by Reviewer x6tG.
>
> > **W1:** Since this paper combines many techniques such as skill-conditioning in a prompt, prompt refinements by improvement tips, and code-based policy, it is rather hard what is the main contribution. If the main contribution is to improve a base prompt by using textual tips, how does O3D differ from recent works like Reflexion?
>
> The main contribution of this paper is to propose a generic approach, O3D, that learns from offline data without updating model weights. Please note that O3D is essentially different from recent works like Reflexion due to the following reasons.
> - Reflexion lets the LLM agent reflect based on its online interaction history, and it takes multiple interaction trials to achieve improved performance. However, in practice, online interaction with the real environment can be expensive or risky. In contrast to Reflexion, our O3D learns from offline data, which can be the interaction log from any other users or agents. O3D achieves improvement in the first trial of deployment without extra online interactions.
> - More importantly, Reflexion learns task-specific textual tips, which only works for the current task it is solving. For example, the tip can be “in the next trial, I should put the apple on the countertop.” Such a tip cannot be directly transferred to a new task with a different goal, and the reflective text needs to be generated for each downstream task. In contrast, O3D discovers reusable skills and generalizable textual tips. For example, the tip can be “I should strictly follow the task instruction and put the object in the desired location.” Such tips facilitate a wide range of downstream tasks in the task distribution (refer to Fig.2 and Section 4.2)
> - Also, Reflexion relies on online failure experiences, however, our approach offers a way to leverage both successful and failed experiences for improvements, which is more effective to generate proper tips to fix mistakes.
> - Last but not the least, our offline learning framework can benefit both text-based policies and code-based policies, leading to wider applicability in real-world domains. Note that developing a method that reliably helps both solutions is very challenging due to the different nature of text generation and code generation.
>
> > **W2:** One of the main concerns on this paper is experiment. Since this paper compares O3D with only one baseline (i.g., ReAct for text-based policy and Demo2Cod for code-based policy), it is hard to properly assess the performance. For example, there are more recent works such as Reflexion for text-based policy. It is highly required to add more recent works and properly compare O3D with them. Also, the ReAct paper provides its performance by using PaLM and text-davinci-002 (GPT-3). However, this paper provides results based on GPT-4 and GPT-3.5. This mismatch makes the comparison more difficult.
>
> Thank you for the suggestion. We would like to first emphasize that we are the first to leverage offline data for sequential decision making without model training. Regarding this objective, there does not exist many baselines to compare with. To address the reviewer's concerns, we conducted extra experiments according to the suggestion. Due to time limitations, not all GPT models are used. We will put the complete results across multiple LLM models in the final version of the paper.

---

> ### Author Response · Authors · 2023-11-21
> **[2 / 3] Response to Reviewer x6tG**
>
> **1. Comparison with Additional Baselines (Reflexion and AdaPlanner)**
>
> *Table A:* Comparison of Text-based-Policy with Reflexion on ALFWorld with GPT-3.5-0613 (success rate in %). Reflexion is reproduced from the original codebase, with 10 trials (original setting in the paper).
> | Method | Pick   | Clean  | Heat   | Cool   | Look   | Pick2  | ALL |
> | :----: | :-----: | :----: | :----: | :----: | :----: | :----: | :----: |
> |Relfexion | 33 | 26 | **26** | 24 | **50** | 18 | 29 |
> |O3D       | **71** | **35** | 4 | **67** | 44 | **24** | **41** |
> |||||||
>
> *Table B:* Comparison of Text-based-Policy with Reflexion on WebShop with GPT-4. Reflexion is reproduced from the original codebase, with 3 trials (original setting in the paper).
> | Method | Success Rate   | Score  |
> | :----: | :-----: | :----: |
> |Relfexion | 0.27 | 0.31 |
> |O3D       | **0.41** | **0.58** |
> |||
>
> *Table C:* Comparison of Code-based-Policy with AdaPlanner[https://arxiv.org/abs/2305.16653] on ALFWorld with GPT-3.5-0613 and GPT-3.5-0301 (success rate in %). AdaPlanner is reproduced from the original codebase[https://github.com/haotiansun14/AdaPlanner]. GPT-3.5-0301 results are provided by the AdaPlanner paper.
>
> *Table C.1.* Comparison Results with GPT-3.5-0613.
> | Method | Pick   | Clean  | Heat   | Cool   | Look   | Pick2  | ALL |
> | :----: | :-----: | :----: | :----: | :----: | :----: | :----: | :----: |
> |AdaPlanner | 4 | 13 | 0 | 0 | 0 | 0 | 13 |
> |O3D-code | **100** | **71** | **91** | **86** | **89** | **18** | **78** |
> ||||||
>
> *Table C.2.* Comparison Results with GPT-3.5-0301.
> | Method | Pick   | Clean  | Heat   | Cool   | Look   | Pick2  | ALL |
> | :----: | :-----: | :----: | :----: | :----: | :----: | :----: | :----: |
> |AdaPlanner | 78 | **94** | 70 | **94** | 63 | 78 | 81 |
> |O3D-code | **96** | 84 | **91** | 86 | **89** | **88** | **89** |
> ||||||
>
> Note that AdaPlanner does not provide the solution for WebShop, thus we cannot compare with it. As the results shown above, O3D-code consistently outperforms AdaPlanner over the two GPT models.
>
>
> **2. O3D Can Be Combined with Existing Methods to Enhance Their Performance**
>
> It is important to note that our offline learning framework is orthogonal to many baseline methods which aims to improve the few-shot learning or online learning of LLM agents. Therefore, our method can be combined with these methods to enhance them, by prepending the distilled tips and discovered primitives from offline data to the downstream prompt. (For simplicity, one can assume there is only one skill that solves the task, such that we only need to distill tips and primitives for the main policy and add them to the downstream prompt.) Below we show the comparison between Reflexion and Reflexion + O3D.
>
> *Table D:* Combining O3D and Reflexion in ALFWorld with GPT-3.5-0613.
> | Method | Pick   | Clean  | Heat   | Cool   | Look   | Pick2  | ALL |
> | :----: | :-----: | :----: | :----: | :----: | :----: | :----: | :----: |
> |Relfexion | 33 | 26 | 26 | 24 | 50 | 18 | 29 |
> |Reflexion + O3D | **54** | **35** | **61**| **67** | **56**| **29**| **50**|
> ||||||
>
> *Table E:* Combining O3D and Reflexion in WebShop with GPT-4.
> | Method | Success Rate   | Score  |
> | :----: | :-----: | :----: |
> |Relfexion | 0.27 | 0.31 |
> |Reflexion + O3D | **0.31** | **0.37** |
> |||||
>
> From the above results, we can see that O3D can significantly improve the performance of Reflexion, which further demonstrates that our offline discovery and distillation idea is generalizable such that it can advance any existing approaches.
>
> **3.Additional Results with GPT-3**
>
> To address the reviewer's concerns, we show extra results in ALFworld using GPT-3 (text-davinci-002) below. Meanwhile, we would like to mention that, the objective of our work is not to compare the best performance over different models, but to show that our approach can consistently outperform baselines using any existing models.
>
>
> *Table F:* Comparison between ReAct, O3D and O3D-Human with GPT-3 (appending to Table 1 in paper).
> | Method | Pick   | Clean  | Heat   | Cool   | Look   | Pick2  | ALL |
> | :----: | :-----: | :----: | :----: | :----: | :----: | :----: | :----: |
> |ReAct | 67 | 65 | 61 | 86 | 78 | 53 | 68 |
> |O3D   | **83** | **97** | **78** | **86** | 83 | **76** | **85** |
> |O3D-Human  | 79 | 97 | 74 | 86 | **94** | 71 | 84 |
> ||||||
>
> The above results show that our method still outperforms ReAct with GPT-3, consistent with the observation in paper with other models. For WebShop, the ReAct paper does not provide the GPT-3 results, and the codebase for PaLM are not provided, either. We also do not have access to the PaLM models used in ReAct. Therefore, it is not feasible to compare in this setting.

---

> ### Author Response · Authors · 2023-11-21
> **[3 / 3] Response to Reviewer x6tG**
>
> > **W3:** According to Table 1 and 2, code-based policy does not work well in WebShop. This result does not seem to support that O3D consistently outperforms baselines in both text-based policy and code-based policy across various LLMs.
>
>
> We would like to mention that, this is the first work to investigate the pros and cons of text-based policy and code-based policy in the same domain. We found that some domain naturally fits code-based policy (like ALFWorld), but some domain naturally fits text-based policy (like Webshop). Even in the case of Webshop where code-based policy is not preferable, we have shown O3D outperforms the baselines Demo2Code on GPT-4. Regarding the equally poor performance on GPT-3.5-0613, this is mainly caused by the model's coding limitation under WebShop domain.
>
>
> Additional experiments in Table C above show that our O3D-Code achieves good performance with GPT-3.5-0301, and it also significantly outperforms another recent code-based approach AdaPlanner, which further verifies the advantage of O3D for Code-based-Policy.
>
>
>
> > **W4:** The proposed method seems more suitable for ALFWorld than WebShop. I am not sure that O3D is generally applicable to diverse environments.
>
> 1. WebShop is a challenging domain, where ReAct and Reflexion also do not achieve high scores, and code-based baselines do not even attempt to solve this domain. Instead, our O3D **outperforms ReAct and Reflexion** (**by a large margin with the GPT-4**, the SOTA LLM model) and achieves **non-trivial progress** in code-based policies. It already shows the advantage of O3D in solving the WebShop domain. Our additional experiments in the response to W2 above further verify the effectiveness of O3D in WebShop.
> 2. ALFWorld and WebShop are commonly used benchmarks of text-based sequential decision making. These two domains are very different, and are representative for many real-world domains covering robot manipulation, navigation, web interaction, e-commerce, etc. We emphasize that O3D beats the baselines in both domains with both text-based-policy and code-based-policy, which demonstrates the wide applicability of O3D.
> 3. The main idea of O3D is to discover and distill reusable knowledge from offline data, which mimics how humans learn from offline experience. This idea is meant to be generic and not specific to any domain. Section 4.2 shows the detailed analysis for the distilled knowledge, which aligns well with human-designed knowledge. It is well known that prompting LLMs with better domain-relevant knowledge can improve its performance. Therefore, the LLM-distilled knowledge should be able to benefit various LLM agents.
>
>
> > **Q1.** Regarding W1, how does O3D differ from self-refinement works like Reflexion?
>
> Please see our response to **W1** above.
>
> > **Q2** How does the skill discovery work for webshop?
>
> For WebShop, we give an example that segment and discover skill from a **partial trajectory** that only searches and selects an item. Given this simple example, we iterate over all the 95 offline trajectories and ask the LLM to discover new skills and segment trajectories. The LLM finally finds 4 skills: search_item(), select_item(), select_item_attributes(), purchase_item(). The prompt used is shown in Appendix B.1.1.
>
>
> > **Q3** What are the details of O3D-human?
>
> O3D-human is designed as an oracle, where a domain expert writes tips and primitives based on the interaction logs of ReAct on testing tasks in both domains. Please see Appendix B.3 for the full human-generated tips and primitives for different skills under each domain, as a comparison, the LLM-generated tips and primitives can be found in Appendix B.4.
>
>
> ---
> We hope our answers and results above have addressed the concerns of Reviewer x6tG. If so, we would greatly appreciate it if Reviewer x6tG could consider raising the score. Please let us know if there are any other questions.
>
> Paper 8704 Authors

---

### Official Review · Reviewer_TtDd · 2023-11-04

**Soundness:** 3 good
**Presentation:** 3 good
**Contribution:** 3 good
**Rating:** 6
**Confidence:** 3

**Summary:**

This paper proposes an offline in-context learning framework. It composes three main steps: 1). the agent learn from a large set of offline logs to segment and discover sub-tasks; also generate various skills/sub-tasks based dataset; 2). from the offline dataset, it asks to model to summarize valid primitive actions, and using trajectory contrasting for generating important tips; All of these will be used as prompts for sub-skill policies; 3). it utilizes few shot prompting to compose these policies and interact with the downstream task.

**Strengths:**

- The paper proposes an interesting idea about how to effectively distill the knowledge from offline datasets into LLM, w.o. expensive finetuing. ICL can only distill several examples through few-shot due to limited context length. The idea of decomposing tasks, learning task-specific prompts via summarizing primitive actions and extracting valid tips using offline dataset to improve the policy is interesting.

- The proposed method is also validated empirically via its superior performance on ALFWorld, WebShop, compared with other multi-step reasoning methods, such as ReAct.

- The paper is very well-written and easy to understand.

**Weaknesses:**

- One thing is not super clear is the effectiveness of each stage of the pipeline. For example, in the skill discovery skill, give the huge amount of offline dataset, probably with noise, there might be hallucinated skills, how this approach effectively handles this and selects the important/valid skills that will be used in downstream tasks. Beyond this, in the second stage, how is the LLM's performance is doing the sub-trajectory extraction? It would be great to provide this more fine-grained analysis to understand the limitations and effectiveness of different stages.

- Another question that I am curious is that how is the trajectory contrasting compared with RLHF? Given a fixed set of contrasting trajectories, it would be more effective to prompting the models to extract general tips, or it would be great to distill the knowledge using RLHF?

- For stage 2, how many samples are used to generate primitives as well as distilled tips? And how the positive and negative examples sampled in learning the tips?

- Could you comment more on the effectiveness of self-learned tips generated using Step 2 and human-generated principles? It would be great to list some scenarios on where the self-learned tips is much better.

**Questions:**

See Weakness.

---

> ### Author Response · Authors · 2023-11-21
> **[1 / 2] Response to Reviewer TtDd**
>
> We greatly appreciate the valuable feedback and positive comments from Reviewer TtDd. We are encouraged that the reviewer finds our ideas interesting, the performance superior, and the paper well-written. Below we address the concerns and questions raised by Reviewer TtDd.
>
> > **W1:** One thing is not super clear is the effectiveness of each stage of the pipeline. For example, in the skill discovery skill, give the huge amount of offline dataset, probably with noise, there might be hallucinated skills, how this approach effectively handles this and selects the important/valid skills that will be used in downstream tasks. Beyond this, in the second stage, how is the LLM's performance is doing the sub-trajectory extraction? It would be great to provide this more fine-grained analysis to understand the limitations and effectiveness of different stages.
>
> Thank you for the question. We address it by providing a detailed analysis of how to ensure the effectiveness of Stage 1 and 2 below. We then illustrate the potential limitations in the process.
>
> 1. *Analysis of Stage 1: skill discovery and segmentation.*
>
> - In this stage, we ask the LLM to segment given trajectories and label each sub-trajectory as some skill. To improve accuracy, one demonstration is provided in the prompt (refer to Appendix B.1.1) to guide LLMs to discover the desired skills in a particular format, such as segmenting the trajectory and labeling each sub-trajectory as a skill.  For example, in ALFWorld, we provide one demonstration on a trajectory of one "Pick" task, then the LLM can imitate it (in-context learning) to generate the segmentation with skill labels Across 6 types of tasks, and eventually output a list of skills.(For Webshop, please check the prompt details in Appendix B.1.1)
> - As the discovered skills are grounded on factual trajectories, with the demonstration, we observe stable and high-quality skill discovery results without hallucination. Variants of skill names may exist, while we instruct LLMs by saying something like not only appending new skills but also performing refinement over iterations to avoid duplicate skills in the final output.
>
> 2. *Analysis of Stage 2: primitive discovery and knowledge distillation.*
> - In this stage, we also ask the LLM to refine the discovered primitives and distilled tips over iterations in prompts. (refer to Appendix B.1.2 and B.1.3.) As a result, even if LLM happens to distill some biased tips, it is able to correct it after seeing new data.
> - Our proposed trajectory contrasting can encourage the LLM to pay attention to the difference between a successful trial and a failed trial. Since we sample the (success, and failure) pair from the same task, it is easy for LLM to figure out the failure reason and improvement tips, rather than "hallucinate" irrelevant tips.
> - In the case where some sub-trajectory segmentations from Stage 1 are not accurate, such as assigning some state-action pairs to another skill instead of the right one, Stage 2 may distill some tips or primitives that is not relevant to the current skill. But it does not have a major influence on the performance, because the distilled tips/primitives in this case are **redundant instead of wrong**. Also, the iterative refinement can filter out some irrelevant information.
>
> 3. *Potential Limitations.*
> - Compared to human-provided domain knowledge, O3D uses LLMs to distill domain knowledge from data. Although it significantly saves human efforts, the distilled knowledge is not guaranteed to be thorough and correct. Therefore, we suggest to monitor and inspect the outcome of each stage to ensure high performance in practical usage.
>
>
> > **W2:** Another question that I am curious is that how is the trajectory contrasting compared with RLHF?
>
> This is a very interesting point. We emphasize the following differences and relations.
> - During the RLHF process, the model's weights have to be updated, which is a totally different line of research from our work, as we are exploring the best way to leverage offline data without fine-tuning the model's weights. With the notations defined in Section 3.1, RLHF updates $\theta_{pret}$, while O3D updates $\theta_{pmt}$, both using offline data.
> - The high-level idea of Trajectory Contrasting is analogous to the preference model of RLHF, where a desired behavior is encouraged and an undesired behavior is penalized. RLHF learns it by training a reward model to guide policy learning, but O3D directly distills such preferences as "policy improvement tips".

---

> ### Author Response · Authors · 2023-11-21
> **[2 / 2] Response to Reviewer TtDd**
>
> > **W3:** For stage 2, how many samples are used to generate primitives as well as distilled tips? And how the positive and negative examples sampled in learning the tips?
>
> Please see Appendix A.1 for all details, where we have posted the number of trajectory samples for O3D and the ablation studies.
>
> - In ALFWorld, we use 12 trajectories for primitives discovery and 80 pairs of success and failure trajectories for tips distillation.
> - In Webshop, we use 90 trajectories for primitives discovery and 30 pairs of success and failure trajectories for tips distillation.
>
> The positive samples are human demonstrations provided on the official ALFWorld and Webshop repository. The negative samples are generated by running ReAct. As we mentioned above, the (success, failure) pair is sampled from the same task, which facilitates LLM to figure out the failure reason and improvement tips, rather than “hallucinate” irrelevant tips.
>
> > **W4:** Could you comment more on the effectiveness of self-learned tips generated using Step 2 and human-generated principles? It would be great to list some scenarios on where the self-learned tips is much better.
>
> We would like to emphasize that O3D-Human is designed as an oracle, where a domain expert writes tips and primitives in the prompt. Given the same self-learned tips and human-generated tips to three different models, as the results show in Table 1&2, which kind of tip is better actually depends on the model's capability, such as in AFLWorld, O3D outperforms O3D-Human on GPT-4 but is worse than O3D-Human on another two GPT-3.5 models.
>
> We noticed that **human often gives more general tips/primitives, while self-learned can distill both general tips/primitives and granulated ones** because of the contrastive way over success-failure pairs. Please see some scenarios below (refer to Appendix B.3 and B.4 for the full tips and primitives of human-generated and LLM self-learned):
>
>
> - On the "Put" skill in ALFWorld, we list several self-learned tips that seem better than human-generated ones, which could be the reason why O3D always outperformed O3D-Human on the "Pick" task across three GPT models.
>
>
> | Method | Tips |
> | :----: | :-----|
> |LLM Self-learned | 1. Use the correct preposition when placing an object on a surface. The correct action is to put the object "in/on" the surface, not just "on" the surface. 2. Place the object in the first available and suitable location instead of unnecessarily checking other locations. 3. If you put an object in a device (like a microwave), remember to take it out before trying to put it in another location.
> |Human-generated | 1. You can directly put an object in/on an occupied receptacle. 2. Please strictly follow the syntax of primitives. |
> |||
>
> - In Webshop, we show a comparison of the tips and primitives of select_item_attributes, where the self-learned way seems doing a better job, so that O3D can outperform O3D-Human on GPT-3.5-0613.
>
> | Method | Tips | Primitives |
> | :----: | :-----| :-----|
> |LLM Self-learned |1. Ensure to select all the necessary attributes as per the instruction. 2. Be aware that color and size options may not always be straightforward and may include additional information or codes. 3. Be careful with the case and spacing of the words when selecting item attributes. | click[Attribute]; click[Color]; click[Style]; click[Size]; click[Flavor]
> |Human-generated | 1. Remember to select all required attributes using click[Attributes] before click[Buy Now]; 2. You should pay attention to the current observation to check clickable buttons; |  click[Attribute] |
> |||
>
> Overall, the comparison between O3D and O3D-Human verifies that LLMs can distill tips and discover primitives in Stage 2 as good as human-generated ones. O3D, thus, is competitive with and sometimes surpasses O3D-Human.
>
> ---
> We hope our answers and results above have addressed the concerns of Reviewer TtDd. Please let us know if there are any other questions.
>
> Paper 8704 Authors

---

### Official Review · Reviewer_UZ4F · 2023-11-09

**Soundness:** 3 good
**Presentation:** 3 good
**Contribution:** 2 fair
**Rating:** 6
**Confidence:** 3

**Summary:**

The paper proposes a three-stage approach for offline policy improvement with knowledge distillation of LLMs through discovering reusable skills and distilling generalizable knowledge across multiple tasks based on offline interaction data.
The experiment uses three GPT models and official human demonstrations as success data, and a set of failure data generated by using ReAct on the training task set introduced in the original ALFWorld and WebShop implementations. The results show that the O3D framework outperforms baseline methods in two challenging benchmark domains, and achieves few-shot generalization to various types of tasks with a single set of prompts.

**Strengths:**

1. Unlike simple behavior cloning of expert trajectories in certain scenarios, the paper proposes a novel three-stage approach for offline policy improvement with knowledge distillation, which can improve the performance of large language models in solving complex and long-horizon tasks.
2. The experiments show improvement compared to previous baselines, the authors also conducted ablation studies to show the effectiveness of each stage in the proposed method.

**Weaknesses:**

1. The paper only compared with ReAct on textual action approaches, while the offline dataset is collected by sampling trajectories with ReAct. Hence, it's reasonable to see significant improvement given the proposed knowledge/skill distillation procedure. I'm wondering if there are similar approaches that could take successful/failed experiences into LLM's memory to improve the policy itself, for example, Reflexion(https://arxiv.org/pdf/2303.11366.pdf), will you compare your method with it?
I also have similar concerns about lacking baselines on code generation approaches.
2. The O3D Human baseline sounds tricky since humans can give almost as good knowledge of skills and primitives as LLMs'.
3. I'm not sure why unifying textual action generation and action with code generation can be a contribution, can you show me why this is of great importance and what is the challenges of unifying these two approaches?

**Questions:**

1. As the author mentioned: 'The proposed framework unifies two common approaches of LLM-based decision-making which is textual action generation and code generation.', I'm not very clear why unifying textual and code action generation is important, though the authors mentioned the advantages of both generation formats.
2. In the pseudocode of algorithms 2 and 3, I'm wondering how to implement the 'segment process' in order to segment them based on skills, is this process also done by the LLMs?
3. Why in Alfworld GPT3.5-0613 is much worse than GPT3.5-0301?
4. For 'Trajectory Contrasting', how to choose trajectories for the LLMs to compare, do you pick similar trajectories or randomly sample the trajectories to compare?
5.  In this paper, the authors argue that one of reasons that skill distillation and tip distillation are important is due to the lack of context length, since the experiments are conducted with GPT4. I think maybe some experiments with GPT4-32k should be conducted to see what will happen when only prepending sufficient long history in the prompt and comparing the results with the proposed framework.

---

> ### Author Response · Authors · 2023-11-21
> **[1 / 4] Response to Reviewer UZ4F**
>
> We greatly appreciate the valuable feedback of Reviewer UZ4F. We are encouraged that the reviewer finds our approach novel and effective. Below we address the concerns and questions raised by Reviewer UZ4F.
>
> > **W1: Baselines.** I'm wondering if there are similar approaches that could take successful/failed experiences into LLM's memory to improve the policy itself, for example, Reflexion(https://arxiv.org/pdf/2303.11366.pdf), will you compare your method with it? I also have similar concerns about lacking baselines on code generation approaches.
>
> Comparing with Reflexion, we would like to first highlight several major differences:
>
> - Reflexion lets the LLM agent reflect based on its online interaction history, and it takes multiple interaction trials to achieve improved performance. However, in practice, online interaction with the real environment can be expensive or risky. In contrast to Reflexion, our O3D learns from offline data, which can be the interaction log from any other users or agents. O3D achieves improvement in the first trial of deployment without extra online interactions.
> - More importantly, Reflexion learns task-specific textual tips, which only works for the current task it is solving. For example, the tip can be “in the next trial, I should put the apple on the countertop.” Such a tip cannot be directly transferred to a new task with a different goal, and the reflective text needs to be generated for each downstream task. In contrast, O3D discovers reusable skills and generalizable textual tips. For example, the tip can be “I should strictly follow the task instruction and put the object in the desired location.” Such tips facilitate a wide range of downstream tasks in the task distribution (refer to Fig.2 and Section 4.2)
> - Also, Reflexion relies on online failure experiences, however, our approach offers a way to leverage both successful and failed experiences for improvements, which is more effective to generate proper tips to fix mistakes.
>
> To address the reviewer's concerns and better evaluate our proposed method, we conducted extra experiments according to the suggestion (see below). Due to time limitations, not all GPT models are used. We will put the complete results in the final version of the paper.

---

> ### Author Response · Authors · 2023-11-21
> **[2 / 4] Response to Reviewer UZ4F**
>
> **1. Comparison with Additional Baselines (Reflexion and AdaPlanner)**
>
> *Table A:* Comparison of Text-based-Policy with Reflexion on ALFWorld with GPT-3.5-0613 (success rate in %). Reflexion is reproduced from the original codebase[https://github.com/noahshinn024/reflexion], with 10 trials (original setting in the paper).
>
> | Method | Pick   | Clean  | Heat   | Cool   | Look   | Pick2  | ALL |
> | :----: | :-----: | :----: | :----: | :----: | :----: | :----: | :----: |
> |Relfexion | 33 | 26 | **26** | 24 | **50** | 18 | 29 |
> |O3D       | **71** | **35** | 4 | **67** | 44 | **24** | **41** |
> |||||||
>
> *Table B:* Comparison of Text-based-Policy with Reflexion on WebShop with GPT-4. Reflexion is reproduced from the original codebase, with 3 trials (original setting in the paper).
> | Method | Success Rate   | Score  |
> | :----: | :-----: | :----: |
> |Reflexion | 0.27 | 0.31 |
> |O3D       | **0.41** | **0.58** |
> |||
>
> *Table C:* Comparison of Code-based-Policy with AdaPlanner[https://arxiv.org/abs/2305.16653] on ALFWorld with GPT-3.5-0613 and GPT-3.5-0301 (success rate in %). AdaPlanner is reproduced from the original codebase[https://github.com/haotiansun14/AdaPlanner]. GPT-3.5-0301 results are provided by the AdaPlanner paper.
>
> *Table C.1.* Comparison Results with GPT-3.5-0613.
> | Method | Pick   | Clean  | Heat   | Cool   | Look   | Pick2  | ALL |
> | :----: | :-----: | :----: | :----: | :----: | :----: | :----: | :----: |
> |AdaPlanner | 4 | 13 | 0 | 0 | 0 | 0 | 13 |
> |O3D-code | **100** | **71** | **91** | **86** | **89** | **18** | **78** |
> ||||||
>
> *Table C.2.* Comparison Results with GPT-3.5-0301.
> | Method | Pick   | Clean  | Heat   | Cool   | Look   | Pick2  | ALL |
> | :----: | :-----: | :----: | :----: | :----: | :----: | :----: | :----: |
> |AdaPlanner | 78 | **94** | 70 | **94** | 63 | 78 | 81 |
> |O3D-code | **96** | 84 | **91** | 86 | **89** | **88** | **89** |
> ||||||
>
> Note that AdaPlanner does not provide a solution for WebShop, thus we cannot compare with it.
>
> As the above results show, O3D and O3D-code outperform Reflexion and AdaPlanner over multiple domains and GPT models, respectively.
>
>
>
> **2. O3D Can Be Combined with Existing Methods to Enhance Their Performance**
>
> It is important to note that our offline learning framework is orthogonal to many baseline methods which aim to improve the few-shot learning or online learning of LLM agents. Therefore, our method can be combined with these methods to enhance them, by prepending the distilled tips and discovered primitives from offline data to the downstream prompt. (For simplicity, one can assume there is only one skill that solves the task, such that we only need to distill tips and primitives for the main policy and add them to the downstream prompt.) Below we show the comparison between Reflexion and Reflexion + O3D.
>
> *Table D:* Combining O3D and Reflexion in ALFWorld with GPT-3.5-0613.
> | Method | Pick   | Clean  | Heat   | Cool   | Look   | Pick2  | ALL |
> | :----: | :-----: | :----: | :----: | :----: | :----: | :----: | :----: |
> |Relfexion | 33 | 26 | 26 | 24 | 50 | 18 | 29 |
> |Reflexion + O3D | **54** | **35** | **61**| **67** | **56**| **29**| **50**|
> ||||||
>
> *Table E:* Combining O3D and Reflexion in WebShop with GPT-4.
> | Method | Success Rate   | Score  |
> | :----: | :-----: | :----: |
> |Relfexion | 0.27 | 0.31 |
> |Reflexion + O3D | **0.31** | **0.37** |
> |||
>
> From the above results, we can see that O3D can significantly improve the performance of Reflexion, which further demonstrates that our offline discovery and distillation idea is generalizable and can actually advance existing approaches.

---

> ### Author Response · Authors · 2023-11-21
> **[3 / 4] Response to Reviewer UZ4F**
>
> > **W2: About O3D-Human.** The O3D Human baseline sounds tricky since humans can give almost as good knowledge of skills and primitives as LLMs'.
>
>
> 1. We would like to emphasize that O3D-Human is designed as an **oracle**, where a domain expert writes tips and primitives in the prompt. Such human knowledge is of high quality and can improve the decision-making performance a lot. In contrast, O3D uses LLMs to distill domain knowledge from data. Therefore, the comparison between O3D and O3D-Human verifies whether LLMs can distill knowledge in Stage 2 that is as good as humans.
> 2. Our experiment shows that O3D is comparable with and sometimes surpasses O3D-Human. This is a promising result as it shows the noticeable ability of LLMs to automatically distill knowledge from offline data.
> 3. In practice, it can be hard and expensive to let humans write appropriate prompts with domain knowledge. Instead, O3D can learn such domain knowledge from data, which largely saves human effort and cost.
>
>
>
> > **W3: Unifying Textual and Code Policies.** I'm not sure why unifying textual action generation and action with code generation can be a contribution, can you show me why this is of great importance and what is the challenges of unifying these two approaches?
>
> Existing work on LLM agents regards text-based approaches and code-based approaches as two separate branches. A method designed for Text-based-Policy (e.g. ReAct) usually does not work for Code-based-Policy and vice versa. However, as we analyzed in Section 3.4, both approaches have their pros and cons and none of them can dominate the other in all domains. Therefore, it is **important** to improve both approaches in order to generalize to various domains in practice, which is one contribution of our O3D framework.
>
> Developing a method that reliably helps both branches is very **challenging** due to the different natures of text generation and code generation. For text, it is crucial to give proper prompts and avoid hallucination. For code, ensuring logical correctness and reusability are more important.
>
> In this paper, we first provide a formal definition of Text-based-Policy and Code-based-Policy with a single notation system, which makes it easier to consider and compare these approaches. Then, we propose an offline learning framework O3D, which discovers reusable units (skills and primitives) and distills generic policy improvement tips. We show that for both Text-based-Policy and Code-based-Policy, this offline knowledge can significantly boost the performance of the LLM agent.
>
>
> > **Q1:** why unifying textual and code action generation is important, though the authors mentioned the advantages of both generation formats.
>
> Please see our response to W3.
>
> > **Q2:** In the pseudocode of algorithms 2 and 3, I'm wondering how to implement the 'segment process' in order to segment them based on skills, is this process also done by the LLMs?
>
> Yes, this process is done by LLMs. To be more clear, skill discovery and trajectory segmentation are done simultaneously. We ask an LLM to segment a given trajectory into sub-tasks, and label each sub-trajectory as executing a skill. For example, in an ALFWorld task of "clean an apple and place it on the countertop", the LLM should be able to segment the trajectory into 4 pieces: finding an apple, taking an apple, cleaning an apple and putting the apple on the countertop. Then, LLM can summarize 4 high-level skills from them: find(object), take(object), clean(object) and put(object, receptacle).
> The complete prompts we use in both ALFWorld and WebShop are provided in Appendix B.1.1.
>
>
> > **Q3:** Why in Alfworld GPT3.5-0613 is much worse than GPT3.5-0301?
>
> We suspect that the poor performance of GPT-3.5-0613 on ALFWorld domain is because it is fine-tuned with the data and objectives that are diverged from the ALFWorld domain. Similar poor performance on ALFWorld is also reported by the AgentBench paper(https://arxiv.org/pdf/2308.03688.pdf). Despite that GPT-3.5-0613 diverges from ALFWorld, our approach can achieve 34% improvement over ReAct, and improve Reflexion performance by 21% (refer to Reflexion+O3D results).
>
> > **Q4:** For 'Trajectory Contrasting', how to choose trajectories for the LLMs to compare, do you pick similar trajectories or randomly sample the trajectories to compare?
>
> In every iteration, we first randomly sample a task, then sample a successful trajectory and a failed trajectory on this task. This is to help the LLM easily identify the failure reason and the key to success.

---

> ### Author Response · Authors · 2023-11-21
> **[4 / 4] Response to Reviewer UZ4F**
>
> > **Q5:** maybe some experiments with GPT4-32k should be conducted to see what will happen when only prepending sufficient long history in the prompt and comparing the results with the proposed framework.
>
>
> It is a very interesting question. We provide analysis and additional experiments as follows.
> 1. There are indeed models with longer context lengths, but they still have a limit. For example, in WebShop, we have offline logs on 95 tasks, resulting in >90k tokens. Therefore, even GPT4-32K cannot consume all data. Although one may carefully select the most representative trajectory examples to put in the prompt, it is not clear which examples are the most useful for the black-box LLM. It also requires extra human labor and domain knowledge.
> 2. Puting more examples in the prompt can also be much more expensive, because the examples need to be provided for each downstream task. In contrast, we distill a condensed set of tips and primitives, so that we only need a short prompt to solve all downstream tasks.
> 3. As suggested, we prepend the same set of offline trajectories into the original ReAct's prompt as long as we can, with GPT-3.5-turbo-0613-16k and GPT-4-32k on ALFWorld and WebShop, and we name this as ReAct-long. The results are shown in Table F, G and H below. We can see that its performance is better than the original ReAct due to the richer context, but O3D still obtains a higher success rate in both domains. Considering that ReAct-long renders a much higher API cost by having a long prompt, O3D is both more efficient and more effective.
>
> *Table F:* Comparison with ReAct-long using **GPT-3.5-turbo-0613-16k** in ALFWorld tasks.
> | Method | Pick   | Clean  | Heat   | Cool   | Look   | Pick2  | ALL |
> | :----: | :-----: | :----: | :----: | :----: | :----: | :----: | :----: |
> |ReAct | 13 | 10 | 0 | 0 | 17 | 0 | 7|
> |ReAct-long | 71 | 6 | 0 | 5 | 17 | 24 | 21|
> |O3D       | **71** | **35** | **4** | **67** | **44** | **24** | **41**|
> ||||||
>
> *Table G:* Comparison with ReAct-long using **GPT-4-32k** in WebShop tasks.
> | Method | Success Rate   | Score  |
> | :----: | :-----: | :----: |
> |ReAct | 0.26 | 0.39 |
> |ReAct-long | 0.4 | **0.61** |
> |O3D | **0.41** | 0.58 |
> |||
>
> *Table H:* Comparison with ReAct-long using **GPT-3.5-0613-16k** in WebShop tasks.
> | Method | Success Rate   | Score  |
> | :----: | :-----: | :----: |
> |ReAct | 0.27 | 0.60 |
> |ReAct-long | 0.3 | **0.64** |
> |O3D | **0.35** | 0.61 |
> |||
>
> In WebShop, we can see that O3D has a higher success rate with GPT-4 and GPT-3.5, although ReAct-long uses a more powerful version of the model. But we also note that ReAct-long obtains a higher average score than O3D. This is because the current implementation of O3D distills tips by comparing successful trajectories (score=1, when the bought item exactly matches the instruction) and failed trajectories (score<1, when the purchase failed or when the bought item does not exactly match the instruction). As a result, the distilled tips focus on how to achieve an exact match and score=1, instead of increasing the raw score. Since ReAct-long provides the full trajectories with scores, it has a more fine-grained understanding of the relation between tasks and scores. One idea for improving the average score of O3D is to contrast the trajectories with higher scores and lower scores instead of simple successes and failures.
>
>
> ---
> We hope our answers and results above have addressed the concerns of Reviewer UZ4F. If so, we would greatly appreciate it if Reviewer UZ4F could consider raising the score. Please let us know if there are any other questions.
>
> Paper 8704 Authors

---

> > ### Comment · Reviewer_UZ4F · 2023-11-22
> >
> > Thank you for your careful and detailed explanations and experiments. Most of my concerns are addressed after reading your feedback to me and other reviewers.
> >
> > I think the results of Reflexion and Reflexion + O3D on Alfworld-Heat do give me new insights about this method, when simply sampling with GPT-3.5-turbo-0613 by ReAct, the success rate is 0 as shown in Table 1, which makes the performance of O3D weak. However, when the sampling method is Reflexion, since the rollout policy has a better success rate, the successful experience also greatly helps the proposed offline algorithm O3D to boost the performance to a much better one. I guess ReAct-Long can have a similar effect and help O3D boost its performance.
> > Therefore, there are many things to explore, e.g., the exploration/performance/online improving property of the rollout policy.
> > I recommend the author put the results in the paper and discuss under which circumstances the algorithm O3D will work well.
> >
> > About the O3D-human details, I want to ask one more question, does the human expert come up with the tips/skills with the help of an LLM? As you know it is natural now for us humans to summarize the experience and write tips/skills by first asking an LLM and then polish and improve it with our own knowledge.
> >
> > I'm delighted to raise my score.

---

> > > ### Author Response · Authors · 2023-11-22
> > > **Thank you for the reply**
> > >
> > > We greatly appreciate the reviewer’s response, suggestions, and the raised score! The suggestions are very inspiring, and we are happy to explore the proposed directions. We promise to add the above new results along with more discussion into the final version of our paper.
> > >
> > > > I want to ask one more question, does the human expert come up with the tips/skills with the help of an LLM?
> > >
> > > In our experiments, the human expert comes up with the tips/skills **without** any help from an LLM. We do think the reviewer’s proposal is interesting, such that we could certainly explore a version as LLM+human to see if it could achieve extra gain in performance.

---

### Author Response · Authors · 2023-11-21
**General Response**

We greatly appreciate the insightful and high-quality comments from all reviewers. We would like to provide a summary of our new experiments and concisely clarify the contributions of our work below.

**Experiment of Additional Baselines**. Our paper originally demonstrates the outperformance of our approach against ReAct, the SOTA baseline in a wide range of tasks. According to the suggestions of reviewers, we have conducted comparisons against another two baselines:

- Reflexion[1]
- AdaPlanner[2]

We post the results in each individual response, where our approach, O3D, also significantly outperforms the above two baselines.


**Experiments Show That O3D Can Enhance Existing Methods**. We also conducted extra experiments with a comparison between Reflexion and Reflexion+O3D to validate that, our offline learning framework can improve existing methods by integrating the discovered primitives and distilled tips from offline data into their prompts. The results show that O3D significantly boosts Reflexion’s performance under both ALFWorld and Webshop domains.


**Summary of Contribution**

- To our best knowledge, O3D is the first work to improve LLM-agents’ sequential decision-making performance by learning from offline data at scale without fine-tuning the model’s weights.
- O3D can effectively discover skills and distill tips that are generalizable and reusable to unseen downstream tasks.
- To our best knowledge, this is the first work that provides a formal definition of Text-based-Policy and Code-based-Policy with a single notation system. The proposed offline learning framework consistently improves the performance for both types of policy forms, making the method more generic and applicable to various domains.
- Distinct from existing online in-context learning methods with few-shot demonstrations (e.g. ReAct) and self-reflections (e.g. Reflexion), O3D can be used to enhance them with offline data, which offers the possibility of achieving a high-quality policy before deployment and reduces the cost of online interaction.


[1] Noah Shinn, et al., Reflexion: Language Agents with Verbal Reinforcement Learning, NeurIPs, 2023

[2] Haotian Sun, et al., AdaPlanner: Adaptive Planning from Feedback with Language Models, arXiv, 2023

---

### Meta-Review · Area_Chair_oQYG · 2023-12-11

**Metareview:**

This paper proposes an offline learning framework that leverages large-scale offline data, such as human interaction logs, to enhance the in-context learning performance of Language Model-based (LLM) agents. The framework defines LLM-powered policies using text-based and code-based approaches and introduces the Offline Data-driven Discovery and Distillation (O3D) framework. O3D automatically discovers reusable skills and distills generalizable knowledge from offline interaction data, enabling improved performance on downstream tasks without the need for fine-tuning. Some weaknesses or concerns of the paper are raised from the review comments and discussions, including the incremental technical novelty (difference from recent works), discussion of related works, and experiment insufficiency. Although the authors provided detailed feedbacks, some of the concerns raised are still unsolved.

**Justification For Why Not Higher Score:**

Some weaknesses or concerns of the paper are raised from the review comments and discussions, including the incremental technical novelty (difference from recent works), discussion of related works, and experiment insufficiency. Although the authors provided detailed feedbacks, some of the concerns raised are still unsolved.

**Justification For Why Not Lower Score:**

N/A

---

### Decision · Program_Chairs · 2024-01-16

Reject